# Post-Capillary Pulmonary Hypertension: Clinical Review

**DOI:** 10.3390/jcm13020625

**Published:** 2024-01-22

**Authors:** Joshua M. Riley, James J. Fradin, Douglas H. Russ, Eric D. Warner, Yevgeniy Brailovsky, Indranee Rajapreyar

**Affiliations:** 1Department of Medicine, Thomas Jefferson University Hospital, Philadelphia, PA 19147, USA; joshua.riley@jefferson.edu (J.M.R.);; 2Sidney Kimmel Medical College, Thomas Jefferson University, Philadelphia, PA 19147, USA; 3Jefferson Heart Institute, Thomas Jefferson University Hospital, Philadelphia, PA 19147, USA; yevgeniy.brailovsky@jefferson.edu

**Keywords:** pulmonary hypertension, post-capillary, heart failure, advanced heart failure, HFrEF

## Abstract

Pulmonary hypertension (PH) caused by left heart disease, also known as post-capillary PH, is the most common etiology of PH. Left heart disease due to systolic dysfunction or heart failure with preserved ejection fraction, valvular heart disease, and left atrial myopathy due to atrial fibrillation are causes of post-capillary PH. Elevated left-sided filling pressures cause pulmonary venous congestion due to backward transmission of pressures and post-capillary PH. In advanced left-sided heart disease or valvular heart disease, chronic uncontrolled venous congestion may lead to remodeling of the pulmonary arterial system, causing combined pre-capillary and post-capillary PH. The hemodynamic definition of post-capillary PH includes a mean pulmonary arterial pressure > 20 mmHg, pulmonary vascular resistance < 3 Wood units, and pulmonary capillary wedge pressure > 15 mmHg. Echocardiography is important in the identification and management of the underlying cause of post-capillary PH. Management of post-capillary PH is focused on the treatment of the underlying condition. Strategies are geared towards pharmacotherapy and guideline-directed medical therapy for heart failure, surgical or percutaneous management of valvular disorders, and control of modifiable risk factors and comorbid conditions. Referral to centers with advanced heart and pulmonary teams has shown to improve morbidity and mortality. There is emerging interest in the use of targeted agents classically used in pulmonary arterial hypertension, but current data remain limited and conflicting. This review aims to serve as a comprehensive summary of postcapillary PH and its etiologies, pathophysiology, diagnosis, and management, particularly as it pertains to advanced heart failure.

## 1. Introduction

Left heart disease is the most common etiology of pulmonary hypertension (PH) and is associated with poor prognosis. Left heart disease due to systolic dysfunction or heart failure with preserved ejection fraction (HFpEF), valvular heart disease, and left atrial myopathy due to atrial fibrillation are causes of PH (Figure 1). PH due to left heart disease (PH-LHD) can be isolated, resulting in post-capillary PH. In the setting of advanced left-sided heart disease or valvular heart disease, chronic uncontrolled venous congestion may lead to remodeling of the pulmonary arterial system, causing combined pre-capillary and post-capillary pulmonary hypertension [1]. The hemodynamic definition for PH was recently modified. Isolated post-capillary PH was defined as mean pulmonary artery pressure (mPAP) > 20 mmHg, pulmonary arterial wedge pressure (PAWP) > 15 mmHg, and pulmonary vascular resistance (PVR) < 3 Wood units (WU). A subset of patients develop combined pre- and post-capillary PH, which is defined as PVR > 3 WU in combination with elevated mPAP and elevated PAWP [2].

Post-capillary PH can cause right ventricular failure (RVF) due to loss of right ventricle (RV) compensatory mechanisms from chronic RV afterload if the underlying cause of left heart disease is untreated. Systemic venous congestion causes multi-organ failure and is associated with increased mortality and morbidity [3]. In this review, we will discuss the epidemiology, pathophysiology, diagnosis, sequalae, and management of post-capillary PH. 

## 2. Epidemiology

Post-capillary PH is the most prevalent form of PH, estimated at 35–60% of all PH cases [4,5]. The vast majority of post-capillary PH is due to HFpEF and heart failure with reduced ejection fraction (HFrEF). PH is estimated to be present in 40–80% of all cases of HFpEF and HFrEF [6,7]. The morbidity and mortality associated with concomitant PH and left heart disease are profound and dependent on the etiology and severity of the underlying heart disease. The diagnosis of PH in patients with heart failure (HF) is associated with increased mortality [8,9,10]. One study with 307 patients with post-capillary PH, predominantly within New York Health Association Functional Class III, showed 1-, 3-, and 5-year survival rates of 86.7%, 68.6%, and 55.6%, respectively [11]. A Canadian retrospective cohort study of 50,529 patients with a diagnosis of PH found that adults with PH-LHD had 1- and 5-year survival rates of 61.2% and 34.5%, respectively. Prognosis also depends on the absence or development of RVF. In patients who have elevated PAP, reduced RV systolic function is associated with a worse prognosis and poorer survival compared to patients with normal RV function [12]. 

There are no currently identified genetic markers associated with post-capillary PH; however, known genetic markers and polymorphisms associated with cardiomyopathy and valvular dysfunction may help identify those at risk of developing left heart disease leading to post-capillary PH.

## 3. Pathophysiology

The changes in the cardiopulmonary circulation in HFrEF or HFpEF determine the symptoms and prognosis of left heart disease. The left atrium (LA) plays a vital role in the pathophysiology of PH in left heart disease. Left atrial function is divided into three phases as described in the article by Rossi et al.: reservoir, conduit, and booster. The LA has a protective role in buffering the dynamic changes in left-sided filling pressures and resultant mitral regurgitation in the early stages of left ventricular impaired relaxation or contractile dysfunction [13]. Chronic elevation in left atrial pressures (LAP) in HFrEF or HFpEF, or left-sided valvular heart disease, results in LA remodeling and fibrosis with excessive transmission of pressures to the pulmonary circulation [14].

There are differences in LA remodeling and function in patients with HFpEF and HFrEF. In patients with HFpEF, there is evidence of higher peak LAP and LA stiffness despite lower LA dilation and mitral regurgitation severity compared to the HFrEF cohort. In patients with HFrEF, there is left chamber dilation with lower function compared to the HFpEF cohort [15]. There is a higher incidence of atrial fibrillation in the HFpEF cohort, which also perpetuates a vicious cycle of atrial myopathy [15,16]. With HF disease progression, LA compliance and function declines with the loss of the protective role of the LA. The persistent elevation in pulmonary venous pressures leads to the development of pulmonary vascular disease due to pulmonary vascular remodeling. The RV is sensitive to even a modest increase in afterload with greater reductions in stroke volume (SV) compared to the left ventricle (LV) [17]. Elevated PAWP during rest or exercise in HF augments systolic pulmonary artery pressure and decreases pulmonary vascular compliance for a given vascular resistance [18]. Chronically pulsatile loading of the pulmonary vasculature results in alveolar capillary remodeling, endothelial dysfunction, decreased nitric oxide availability, and irreversible pulmonary vascular remodeling. RV afterload increase is transient in the initial stages of HF but may become sustained with the progression of pulmonary vascular remodeling [19]. Short-term RV adaptation (RV–PA coupling) to increased pulsatile load in the pulmonary circuit is not sustained, which leads to progression to the clinical syndrome of RVF (Figure 2). RV–PA uncoupling may be unresponsive to medical or surgical intervention to lower the PA systolic load. In addition, progressive RV dilation due to increased RV afterload can lead to worsening LV function and decreased cardiac output due to pericardial constraint and interventricular dependence [20]. Pulmonary alveolar capillary remodeling and decreased pulmonary perfusion lead to impaired gas exchange and ventilation/perfusion mismatch with symptoms of hypoxemia and exertional dyspnea. Patients who develop pulmonary vascular disease or pre-capillary PH demonstrate an increase in pulmonary congestion with exercise compared to patients with post-capillary PH [21]. 

## 4. Diagnosis

Pulmonary hypertension is defined by the 6th World Symposium on Pulmonary Hypertension as mPAP > 20 mmHg while supine and at rest, correlating with greater than the 97th percentile of average pulmonary pressures in healthy individuals [2,22]. The previous definition that used mPAP > 25 mmHg was modified because studies showed that mortality risk increased by 23% when mPAP was > 19 mmHg [10]. Due to multiple variables that can affect pulmonary arterial pressure, a lone value > 20 mmHg is not always sufficient for diagnosis. Increases in cardiac output (i.e., exercise, left-to-right shunting), hyperviscosity, and elevated PAWP from left heart disease can elevate mPAP in the absence of true pathology within the intrapulmonary vasculature [2]. Further characterization of pre-capillary, post-capillary, and combined pre- and post-capillary PH (cpcPH) is defined by PAWP and PVR: Pre-capillary PH: mPAP > 20 mmHg, PAWP 15 mmHg, PVR > 3 WU
Post-capillary PH: mPAP > 20 mmHg, PAWP > 15 mmHg
CpcPH: mPAP > 20 mmHg, PAWP > 15 mmHg, PVR > 3 WU

Pulmonary vascular resistance is a measure of resistance to blood flow in the pulmonary circulation and is indirectly a measure of RV afterload [23]. Elevated PVR in patients with left heart disease is associated with an increased hazard of mortality and heart failure hospitalizations, and this risk increases with PVR > 2.2 [24]. Even though pulmonary arterial compliance and pulmonary arterial elastance are measures reflective of pulsatile load, they are not routinely used in clinical practice [25].

There are multiple methods of measuring pulmonary pressures and, consequently, multiple ways of detecting pulmonary hypertension (Figure 3). Echocardiography, specifically transthoracic echocardiography (TTE), plays a central role in the initial evaluation of post-capillary PH. TTE allows for the assessment of various hemodynamic parameters, such as left ventricular systolic and diastolic function, valvular abnormalities, and estimates of pulmonary artery pressures. While TTE is an essential tool for diagnosing the underlying cause of left heart disease and post-capillary PH, its estimates of pulmonary artery pressures can be inaccurate in multiple clinical scenarios. The estimation of pulmonary artery systolic pressures (PASP) relies on calculations based on estimates of right atrial pressure and physiologic tricuspid regurgitation. Right atrial pressures are estimated by the size and distensibility of the IVC throughout respiration, which has been shown to greatly underestimate or overestimate the true right atrial pressure [26]. Likewise, absent or severe tricuspid regurgitation can lead to underestimation of PASP and underdiagnosis of PH [27,28]. Additionally, inadequate acoustic windows can further prohibit accurate characterization of pulmonary vasculature pressures. Beyond pulmonary pressure, there are several key echocardiographic parameters that can shed light on the presence and degree of pulmonary vascular disease and RV adaption to increased afterload. The right ventricular outflow tract signal is obtained by placing the pulse wave Doppler proximal to the pulmonic valve. Normally the signal is a parabolic shape spanning almost the entirety of systole. However, in the presence of significant pulmonary vascular disease (pre-capillary PH), the profile becomes notched, with the degree of notching correlating with PVR (mid-systolic notching ~ PVR > 9 WU, late systolic notching ~5 WU) [29]. Similarly, the shape of the RV can elucidate the degree of pulmonary vascular disease. The systolic RV base/apex ratio is significantly lower in patients with pre-capillary PH as compared to those with pulmonary venous hypertension (1.3 vs. 2.6) [30].

Right heart catheterization (RHC) remains the gold standard for diagnosing PH as it provides precise measurements of pulmonary pressures and other hemodynamic parameters [31]. It is an invasive procedure that directly measures the pressure in individual chambers of the right-sided heart and pulmonary vasculature. Pulmonary artery pressures are measured throughout the respiratory and cardiac cycle, yielding systolic, diastolic, and mean pulmonary artery pressure values. As intrathoracic pressures affect the pressures within the pulmonary vasculature, pressure measurements should ideally be recorded at end-expiration.

Additionally, RHC is useful because it can measure PAWP, which provides an indirect measurement of the pressures of the pulmonary venous system and the left atrium. These values are essential in determining if there is a cardiac component associated with PH. RHC can also assess hemodynamic parameters of cardiac output (CO) and index (CI), which should be calculated using the thermodilution method [31,32]. Additionally, pulmonary vascular resistance can be calculated using the equation:$$Pulmonary\;Vascular\;Resistance\;\left(in\;WU\right)=\frac{\left(mPAP-PAWP\right)}{CO}\;$$

While elevated PAWP is classically used as evidence of post-capillary pulmonary hypertension and therefore left heart failure, it is important to note that a normal PAWP does not rule out the diagnosis of HFpEF. Volume challenge or exercise during RHC has been used to attempt to unmask left heart dysfunction in patients with suspected left heart disease and normal PAWP. 

Though RHC is the diagnostic gold standard for PH, the utility of the test is dependent on correct data acquisition and interpretation. Errors in data acquisition can occur in the case of inappropriately calibrated machines and with under- or over-dilation of the catheter balloon [33,34,35]. As mentioned above, the PAWP must be measured at the end of expiration, particularly relevant in the case of lung disease or obesity, where mean PAWP can be significantly more elevated than end-expiration PAWP. Measurement should be taken at the same point of the cardiac cycle and specifically at the “a” wave as mitral regurgitation can cause large “v” waves, impacting mean PAWP [34].

It is also important to note that PAWP is used as a surrogate measurement for left ventricle filling pressure. Another surrogate marker that can be used is left ventricular end-diastolic pressure (LVEDP). LVEDP is most accurately obtained by placing a catheter with a pressure transducer directly in the left ventricle via arterial access. As discussed above with PAWP, echocardiographic estimates of LVEDP have proven unreliable and inaccurate when compared to invasive measurements [36]. Studies have highlighted that PAWP can vary in the accuracy of estimation of pre-load of the left ventricle when compared to LVEDP [37]. While LVEDP may provide a more direct measure of diastolic, and therefore, filling pressure in the LV, PAWP provides additional information regarding the function and adaptability of the left atrium. This insight into LA compensation is likely in part why the use of PAWP, rather than LVEDP, has been shown to have more prognostic utility of morbidity and mortality in patients with heart failure [38].

## 5. Treatment

The management of PH-left heart disease (PH-LHD) is dependent upon the etiology of left heart disease. For this reason, this section will elaborate on the treatments specific to the underlying cardiac etiology of PH. 

### 5.1. Heart Failure with Reduced Ejection Fraction

Managing pulmonary hypertension caused by HFrEF involves a comprehensive approach including optimizing guideline-directed medical therapy (GDMT), addressing fluid balance, and considering targeted therapies for PH. Pharmacotherapy should be initiated in all patients with the goals of reducing mortality, preventing hospitalizations, and improving quality of life [39,40,41]. The four pillars of GDMT include beta-blockers, mineralocorticoid receptor antagonists (MRA), sodium-glucose cotransporter-2 inhibitors (SGLT-2i), and angiotensin-converting enzyme (ACE) inhibitors or angiotensin II receptor blockers (ARB) or angiotensin receptor-neprilysin inhibitors (ARNI). Other medications that can be considered in select patients include loop diuretics, I_f_-channel inhibitors, digoxin, and a combination of hydralazine and isosorbide dinitrate. When indicated, patients with left bundle branch block should also receive cardiac resynchronization therapy [42]. 

An area of recent interest in PH-LHD treatment has been the use of drugs traditionally reserved for PAH in the treatment of PH-LHD. One randomized controlled trial (RCT) comparing bosentan with placebo in patients with HFrEF and PH-LHD showed no measurable hemodynamic benefit and increased adverse events, leading to discontinuation [43]. Another RCT comparing sildenafil with placebo in patients with HFrEF and PH-LHD showed improvements in exercise capacity and quality of life [44]. Another meta-analysis assessing six RCTs involving patients with chronic HFrEF and PH-LHD showed that sildenafil treatment resulted in decreased hospital admissions, reduced mPAP and PVR, increased exercise capacity, and increased quality of life [45]. A trial assessing the soluble guanylate cyclase (sGC) stimulator riociguat in patients with HFrEF and PH-LHD showed improved cardiac index and SVR as compared to the placebo; however, the primary endpoint of a reduction in mPAP was not met [46]. It is important to note that many of these trials have not differentiated between pre-capillary, post-capillary, and CpcPH. Though the results of these small trials show promise for the future use of PAH agents, larger-scale RCTs are needed to further characterize their role in PH-LHD and cpc-PH. The 2022 guidelines from the European Society of Cardiology (ESC) and European Respiratory Society (ERS) do not suggest the use of PAH drugs in patients with PH-LHD [47]. 

### 5.2. Heart Failure with Preserved Ejection Fraction

The management of PH-LHD and HFpEF should be focused on blood pressure control, volume status, and treatment of comorbidities. There is a paucity of evidence suggesting that specific drugs or regimens significantly decrease mortality and morbidity in patients with HFpEF [48,49,50,51,52]. Despite this lack of convincing data, many patients with HFpEF have comorbid hypertension and/or coronary artery disease (CAD) treated with components of GDMT. Guidelines from ESC/ERS suggest treating cardiovascular and non-cardiovascular comorbidities along with symptomatic alleviation of congestion using diuretics [42]. A recent RCT that compared empagliflozin vs. placebo in patients with HFpEF showed a significant reduction in the risk of cardiovascular death and hospitalization for heart failure [53]. 

Similar to HFrEF, there has been interest in PAH drugs and their role in the treatment of patients with HFpEF. Two RCTs investigated the use of endothelin receptor antagonists bosentan and macitentan for patients with HFpEF and PH-LHD, but they did not show significant positive effects and led to more adverse events than the placebo [54,55]. Sildenafil has also been investigated in patients with HFpEF and PH-LHD. Small RCTs have shown that sildenafil does not improve hemodynamics in HFpEF patients with post-capillary PH, while it does improve hemodynamics, RV function, and quality of life in patients with CpcPH [56,57]. Due to the small size of these trials and lack of actionable evidence, there is currently no recommendation for or against the use of sildenafil in patients with HFpEF and CpcPH. However, there are recommendations against the use of sildenafil in patients with HFpEF and post-capillary PH [47].

### 5.3. Advanced Heart Failure

When heart failure is refractory to standard treatment (ACC/AHA Stage D), evaluation for a heart transplant (HT) or mechanical circulatory support is indicated. Elevated PVR > 3 WU is associated with an absolute 1.9% increase in 30-day mortality after HT [58]. The presence of pre-capillary PH in Stage D HF patients can cause severe RVF after HT [59]. Hence, a vasodilator challenge is recommended to determine whether reversibility is performed in patients with elevated pulmonary artery systolic pressure > 50 mmHg and either a transpulmonary gradient > 15 mmHg or PVR > 3 WU. Durable left ventricular assistive devices (LVAD) can be used as bridge-to-transplant or destination therapy for advanced heart failure requiring mechanical circulatory support. LVAD therapy has shown mixed benefits for PH-LHD. In a clinical trial of patients with combined pre- and post-capillary PH (mPAP > 25 mmHg, PVR > 3 WU), implantation of LVAD significantly decreased mPAP and PVR [60]. However, in another study, only approximately one-third of patients experienced normalization of PVR following LVAD [61]. Patients with LVAD support should be monitored for decoupling, defined as a diastolic pulmonary pressure gradient (DPG) of >5 mmHg, which is a predictor of heart failure readmissions and mortality in this population [62].

### 5.4. Preventative Care in Heart Failure

Equally as important as pharmacologic and interventional management of acute and chronic heart failure is the prevention of exacerbations and worsening of preexisting disease. Recommendations for primary prevention of heart failure include treatment of hypertension, hyperlipidemia, and diabetes, as well as counselling for exercise, obesity, tobacco and smoking cessation, and alcohol moderation [63,64,65,66]. Secondary prevention is focused on preventing acute exacerbation in patients with pre-existing heart failure. Current recommendations include appropriate use of the agents listed above in the Section 5.1. These patients should be cared for by a multidisciplinary team managing medication, providing education, and addressing barriers and comorbidities that could increase the risk for morbidity and mortality [67]. All patients with ACC/AHA Stage D heart failure should be referred to an advanced heart failure center.

### 5.5. Valvular Dysfunction

Valvular heart conditions are a common cause of PH-LHD. Definitive treatment including surgical and interventional repair or replacement has shown to improve hemodynamics, but not without frequent persistent PH [68,69]. Since PH-LHD is the result of maladaptive structural changes in the heart chambers and pulmonary vasculature, repair and replacement, while effective in addressing valvular disease, may not immediately reverse PH. The most common valvular disorders to cause PH-LHD include mitral stenosis, mitral regurgitation, and aortic stenosis [70]. 

#### 5.5.1. Mitral Stenosis

Mitral stenosis (MS) causes increased left atrial pressures which elevate pulmonary venous pressures [71]. All patients should receive treatment in line with current guideline recommendations including anticoagulation for select patients with rheumatic disease, heart rate control with beta blockers and calcium channel blockers, and surgical interventions for both rheumatic and calcified MS [72]. There are concerns in the literature regarding operative management of MS in patients with PH. In a study of 317 patients with PH and MS treated with mitral valve surgery, no significant difference in 30-day mortality between different severities of PH was found. Decreased long-term survival in patients with systolic PAP > 45 mmHg was noted [73]. There is still much need for more data regarding these patients and their risk factors to pick the appropriate intervention for treatment of the valvular disorder. A proper pre-operative assessment is essential along with a multidisciplinary team to determine each patient’s risks and benefits with operative management.

#### 5.5.2. Mitral Regurgitation

Mitral regurgitation (MR) leads to elevation of left-sided filling pressures and post-capillary pulmonary pressures [74]. All patients should receive treatment in line with current guideline recommendations including management of hypertension and underlying heart failure, TTE for surveillance of disease severity, and surgical or transcatheter interventions for repair and replacement [72]. Clinicians should be aware that there is recent research detailing the potential risk PH poses in patients pursuing operative correction of MR. Pre-operative PH was associated with reduced post-operative LVEF and increased risk of persistent PH following surgery [75,76]. Mitral valve repair may be preferred in patients with PH, as replacement has been associated with a greater reduction in post-operative LVEF and increased post-operative mortality [76,77]. The findings regarding the risk PH poses in surgical correction of MR are not yet conclusive and more research is needed to identify risk factors for poor operative outcomes. Definitive surgical care should be managed by a team of surgical and medical specialists to determine the best course of action for each individual patient.

#### 5.5.3. Aortic Stenosis

Severe aortic stenosis (AS) causes PH due to compensatory left ventricle hypertrophy leading to increased left atrial pressure transmitted to the pulmonary veins [78]. Recommendations from the ACC/AHA for medical management include standard treatment of hypertension, statin therapy for calcified valves, and ACE-I or ARB for patients who have received transcatheter aortic valve intervention [72]. Aortic valve replacement or repair, whether via a surgical or transcatheter approach, should be performed when indicated; however, clinicians should be aware that pre-intervention PH is associated with increased adverse events and that persistence of PH is common [79,80].

## 6. Conclusions and Future Directions

The development of PH in left heart disease is associated with poor functional capacity and portends a poor prognosis. Management of PH-LHD should be focused on the treatment of underlying etiology and comorbidities. Development of RVF due to chronically elevated RV afterload causes end-organ dysfunction. Prevention of irreversible RVF by management of HF and/or valvular heart disease and early referral to advanced therapies in patients with worsening HF symptoms and persistently elevated PA pressures may improve morbidity and mortality associated with PH.

## Figures and Tables

**Figure 1 jcm-13-00625-f001:**
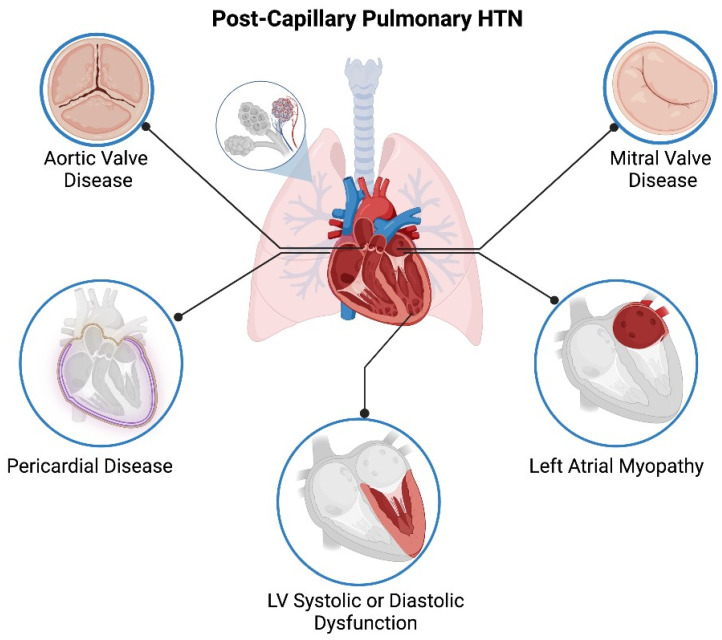
Etiology for Post-capillary Pulmonary Hypertension.

**Figure 2 jcm-13-00625-f002:**
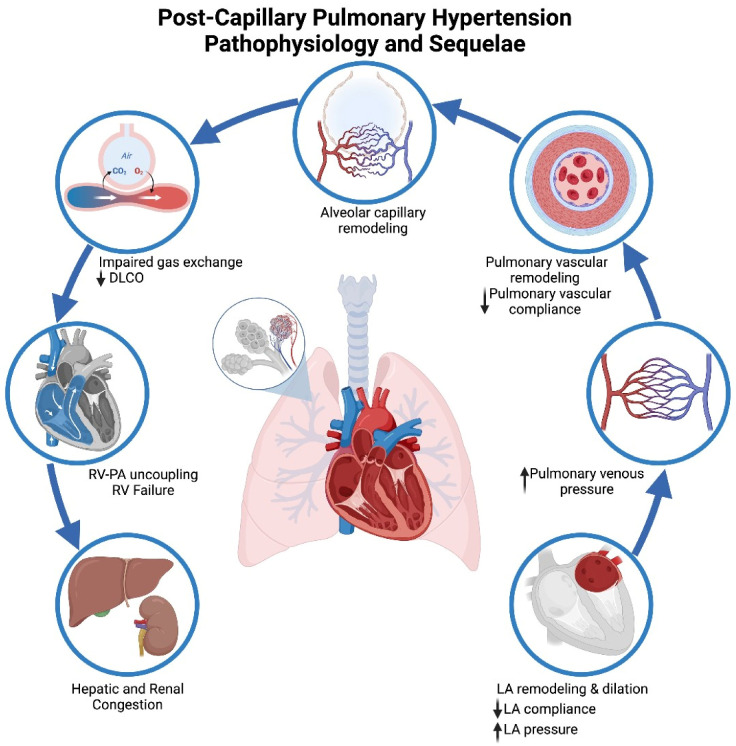
Pathophysiology and Clinical Sequelae of Pulmonary Hypertension due to Left Heart Disease.

**Figure 3 jcm-13-00625-f003:**
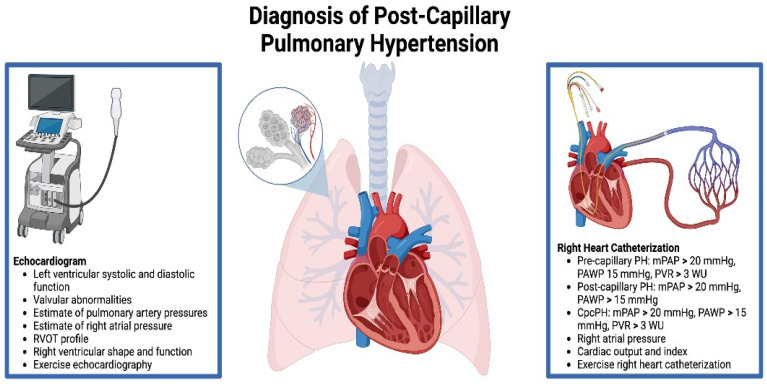
Diagnosis of Post-capillary Pulmonary Hypertension.

## Data Availability

Not applicable.

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
