# Peer review of "Post-Capillary Pulmonary Hypertension: Clinical Review"

_jcm, 2024, doi:10.3390/jcm13020625_

Round 1

Reviewer 1 Report

Comments and Suggestions for Authors

This paper is an interesting narrative review

Paper is well described, according my point of view, one topic is missing

Pulmonary hypertension in COVID-19 survivors ( Pulmonary arterial hypertension and right ventricular systolic dysfunction in COVID-19 survivors, fo example) (I'm not the author)

The figures are precise and important for the readers 

technical sounds are present in the article

A graphical abstract should be important. 

Author Response

Reviewer 1:

Comment: This paper is an interesting narrative review. Paper is well described, according my point of view, one topic is missing: Pulmonary hypertension in COVID-19 survivors ( Pulmonary arterial hypertension and right ventricular systolic dysfunction in COVID-19 survivors, fo example) (I'm not the author). The figures are precise and important for the readers. technical sounds are present in the article. A graphical abstract should be important. 

Response:

Thank you for your feedback on our scholarly work. We agree that the topic of pulmonary hypertension in COVID-19 survivors is interesting, though we intentionally have excluded this from our manuscript. This topic was left out, as the cause of pulmonary arterial hypertension (PAH) in COVID-19 is not fully understood, and largely related to pre-capillary pulmonary hypertension (PH) rather than post-capillary PH. It will be important for future work on pre-capillary PH and PAH to include and consider the role of COVID-19 (and its long-term sequelae) on cardiopulmonary physiology in PH.

If the reviewers and editor would like a graphical abstract, the authors would be happy to combine our Figure 1+2 for a visual representation of our review for JCM to use in publication.

Reviewer 2 Report

Comments and Suggestions for Authors

Good summary. 

Any novel therapies being considered in this area? Any genetic links to this condition? Any biomarkers of relevance?

Author Response

Response:

We appreciate your questions regarding this review. The novel therapies for post-capillary PH are mainly the therapies for PAH discussed in the Treatment section of this review (pg 6; lines 205-222, pg 7; lines 234-244). Unfortunately, at this time, these novel therapies have been met with mixed evidence for use in post-capillary or combined PH.While genetic links have been identified in pulmonary arterial hypertension, no such links have yet been identified in post-capillary hypertension. That being said genetic causes of cardiomyopathy or valvular pathology could be linked to post-capillary PH, if associated with left heart disfunction (Added to p3, para 1). Similarly, though biomarkers are commonly used for PAH, their use in post-capillary PH is limited at this point.

Reviewer 3 Report

Comments and Suggestions for Authors

This is a useful review in particular the section dealing with failure of LA adaptation.

I believe a little more discussion over the diagnostic RHC and measurement of PCW and its problems should be undertaken since it is so crucial in diagnosis but must be done exactly otherwise wrong data is provided. Similarly discussion over the role of LVDP should be greatly expanded 

Author Response

We appreciate your feedback and improvement on our work. The authors have included greater detail about the user-dependence of RHC, as it is vital to the accuracy of the obtained measurements (p6, paragraph 1). Similarly, we have expanded upon our discussion of the LVEDP, a vital measurement in which further future diagnostics of post-capillary PH will likely rely (p 6, paragraph 2).

Round 2

Reviewer 1 Report

Comments and Suggestions for Authors

I accept author reponse